# Rapid Genomic Testing in Intensive Care: Health Professionals’ Perspectives on Ethical Challenges

**DOI:** 10.3390/children10050824

**Published:** 2023-05-01

**Authors:** Katie Arkell, Christopher Gyngell, Zornitza Stark, Danya F. Vears

**Affiliations:** 1Biomedical Ethics Research Group, Murdoch Children’s Research Institute, Parkville, VIC 3052, Australia; katie.arkell@melbournegenomics.org.au (K.A.);; 2Department of Paediatrics, The University of Melbourne, Parkville, VIC 3052, Australia; 3Melbourne Law School, The University of Melbourne, Carlton, VIC 3053, Australia; 4Victorian Clinical Genetics Services, Murdoch Children’s Research Institute, Parkville, VIC 3052, Australia; 5Australian Genomics, Parkville, VIC 3052, Australia

**Keywords:** bioethics, genomic sequencing, NICU, PICU, consent, genetic counseling

## Abstract

Ultra-rapid genomic sequencing (urGS) is increasingly used in neonatal and pediatric intensive care settings (NICU/PICU), demonstrating high diagnostic and clinical utility. This study aimed to explore the perspectives of healthcare professionals (HPs) and the challenges raised by urGS, particularly when making treatment decisions. Four focus groups and two interviews were conducted with HPs who had experience using urGS in NICU/PICU. Inductive content analysis was used to analyze the data. Nineteen HPs participated overall (eight clinical geneticists, nine genetic counselors, and two intensivists). One challenging area of practice identified by HPs was setting realistic expectations for outcomes of urGS among HPs and families. HPs reported modifying pre-test counseling to include life-limiting diagnoses as a possible test outcome and felt concerned about the timing of the test and its impact on parent–child bonding. UrGS results of uncertain prognostic significance posed considerable challenges. Moral distress arose when families and HPs were misaligned regarding treatment goals following the urGS diagnosis. We identified areas of practice that remain ethically challenging for HPs using urGS in the NICU/PICU. HPs experiences of using urGS in the NICU/PICU could inform specialized training in withdrawal of treatment decision making for the genomics workforce.

## 1. Introduction

Ultra-rapid genomic sequencing (urGS) (whole exome or whole genome sequencing completed in <5 days) was specifically developed for use in the neonatal and pediatric intensive care (NICU/PICU) settings to enable swift diagnosis for critically ill patients with underlying genetic conditions. Diagnoses achieved through urGS can be used to guide the clinical management of serious conditions, enable access to life-saving treatment (in some cases within a matter of hours), and provide informed reproductive options for families [1,2,3,4,5]. Even in cases where no treatment is available for a diagnosed condition, the speed at which a diagnosis is obtained can decrease the need for many other low-yield and, at times, invasive tests, prompting redirection of care towards palliation and potentially reducing unnecessary suffering for NICU/PICU patients [4,6,7,8,9]. Although the use of urGS in pediatric and neonatal critical care worldwide has demonstrated clinical benefit, current research indicates that there is still work to be done to understand the ethical concerns that accompany urGS in this setting [1,2,10]. For example, it has been suggested that by using urGS, the time from patient presentation to diagnosis is shorter, which may have a disruptive effect on parent–child bonding and exacerbate moral dilemmas that commonly present in NICU/PICU regarding treatment decisions [3]. Groups in the UK, US, and Australia have explored the impact of using urGS in the NICU/PICU from the parent’s perspective. Survey data collected from families who had experienced urGS suggest there is no evidence bonding was disrupted [11,12,13,14]. However, Bowman-Smart et al. who surveyed parents of NICU/PICU patients in Australia, found that parents can experience shock and have altered thinking about their child after receiving a diagnosis in this setting [13]. 

When considering if urGS exacerbates moral dilemmas regarding treatment decisions, several studies have examined the perceptions and experiences of health professionals (HPs) using urGS in the NICU/PICU. While HPs consistently report perceived clinical utility and low perceived harms, both for diagnostic and uninformative results, [1,2,11] studies with intensive care physicians have reported a preference for genetic service support when using urGS in the NICU/PICU. This suggests that HPs may find making decisions challenging when using this test unaided [15,16].

Assistance and training in the interpretation of data from new technology to support clinical decision making for the treatment of critically ill infants is not novel; Wilkinson discussed the impact on clinical decision making due to the introduction of magnetic resonance imaging scans (MRIs) [17]. As a new technology, urGS is not dissimilar to MRI in that it increases the richness and quantity of data available to help reach a diagnosis, particularly regarding information that may support decisions about treatment goals in the NICU/PICU [17,18]. This ethically challenging aspect of urGS testing remains underinvestigated. As such, we aimed to explore the experiences of HPs using urGS in the NICU/PICU, to understand the ways in which urGS impacts clinical decision making. 

## 2. Materials and Methods

This study utilized a qualitative methodology with focus groups as the primary method of data collection. Purposive sampling was used to approach potential participants within the Australian Genomics Acute Care Genomics (ACG) program, which is evaluating urGS as a first-tier test for diagnosis of critically ill neonatal and pediatric patients with rare disease [2]. Eligible participants were HPs currently working with urGS testing in the NICU/PICU setting. HPs were invited to participate via email through the ACG program. HPs who agreed to participate were assigned to sessions based primarily on their profession and then by availability. Two participants who expressed interest in participating but were unable to attend a scheduled focus group session were offered the option to participate in an interview. 

The focus group format was chosen to enable participants to convey their experiences freely to their peers and allow sensitive subject matter to be discussed in depth [19,20]. As recruitment occurred during COVID-19 lockdowns in Australia, all data collection was done online via ZOOM© [21]. The focus group size was deliberately restricted to a maximum of five participants and two facilitators (KA and DV) to allow each participant’s views to be heard within the time allocated and to comfortably manage the session online. Focus groups were allocated taking into account profession and level of seniority wherever possible. A focus group facilitator guide was developed by the research team with input from the director of the ACG program [22,23,24]. Topics covered included experiences of using urGS to diagnose critically ill children, challenges in making decisions about redirection of care towards palliation, and interactions with other HPs and families in this setting. Due to the sensitive nature of the topic, support service contact details were emailed shortly after each session to participants. Sessions were recorded using ZOOM©, transcribed, and de-identified prior to analysis. 

Inductive content analysis was used to analyze the transcripts, in which codes were developed from the data, rather than deductively [25]. Sections of the transcript were coded into broad categories which were then refined to develop sub-categories. This was managed using NVivo 12 qualitative data analysis software [26]. All transcripts were coded by KA and a subset of the transcripts was co-coded by DV to ensure methodological rigor. Any discrepancies were discussed with the research team until an agreement was reached [27]. The study was approved by The Royal Children’s Hospital, Human Research Ethics Committee, prior to study commencement (ERM Ref: 70281) and all participants provided verbal consent before the focus group discussion/interview commenced.

## 3. Results

### 3.1. Participant Characteristics

Overall, 19 HPs participated in the study (eight clinical geneticists/pediatric specialists; nine genetic counselors; two intensivists), from six different Australian health services. Our analysis identified five categories relating to how urGS impacted HP decision making: (1) prioritization of urGS results; (2) timing of urGS; (3) adaptation of pre-test counseling; (4) differences between HP and family decisions; (5) prognostic uncertainty. Illustrative quotes are presented in tables. Pseudonyms are provided to maintain participants’ anonymity in the following style: session type (focus group (FG) or interview (I)), profession, and participant number. Professions are abbreviated as clinical geneticists (ClinG), pediatric specialists (P), genetic counselors (GC), and intensivists (Int).

### 3.2. Prioritization of urGS Results

HPs observed that the introduction of urGS has created situations where priority is given to obtaining a result from the urGS over other tests. They explained witnessing colleagues wait for the urGS result to be returned before making treatment decisions and expressed concern that this reliance on urGS means clinicians are giving insufficient consideration to the entire clinical picture for each patient when making treatment decisions (Table 1: Quote 1). Some questioned their colleagues’ motivations for seeking a result (Table 1: Quote 2). Tied into this perceived over-reliance, our participants reported seeing some non-genetically trained HPs overstate the chances of getting a diagnosis from urGS when speaking to families (Table 1: Quote 3), which led them to question non-genetically trained HPs’ understanding of the limitations of urGS (Table 1: Quote 4).

### 3.3. Timing of the urGS Test

HPs discussed how the timing of the urGS test in the NICU creates unique challenges. They described situations where they felt providing a diagnosis so early after birth had negatively impacted the parent–child bonding process (Table 2: Quote 1). Some HPs had observed situations where parents had disengaged from having discussions about the next treatment steps for their newborn once their child was given a diagnosis of a severe/life-limiting condition (Table 2: Quote 2). They also mentioned situations where parents had made the decision to redirect care towards palliation, hypothesizing that if the diagnosis had not been found in the first few days of the child’s life the parents would have had more time to bond and may not have made the same decision (Table 2: Quote 3).

### 3.4. Adaptation of Pre-Test Counseling

HPs described the ways in which using urGS in the NICU/PICU setting had changed how they speak to families about the outcomes of the urGS test during pre-test counseling (Table 3: Quotes 1 and 2). They explained that when they first started working in this setting, they would mention that the outcome might be dire but now they spend more time preparing families for this outcome, specifically highlighting the possibility of a life-limiting condition (Table 3: Quote 3). HPs felt that ensuring the family is informed of the full range of outcomes helps make later discussions with families about redirection of care towards palliation easier (Table 3: Quote 4). 

### 3.5. Misalignment of Families and HPs on Redirection of Care towards Palliation

HPs explained that the diagnosis of a condition that impacts quality of life via urGS can create a misalignment between families and treating teams regarding the decision to redirect care towards palliation. HPs described scenarios where the patient’s family did not accept the treatment recommendation of the medical team based on the urGS diagnosis. HPs found it particularly distressing when parents elected to redirect care towards palliation, despite treatment options being available. They discussed experiencing discomfort at ending a life that was potentially worth living (Table 4: Quote 1) and fear that families might regret the decision (Table 4: Quote 2), yet felt the decision is not for them to make and that it ultimately rests with the parents (Table 4: Quotes 3 and 4). 

### 3.6. Prognostic Uncertainty and Decision Making

In some cases, urGS will identify variants for which there is only a handful of previously documented cases; there is no roadmap for predicting likely outcomes and possible treatment options for patients. As described by HPs in our study, obtaining detailed prognostic information assists in predicting outcomes (Table 5: Quote 1). HPs explained how a lack of familiarity with a condition, combined with a paucity of published information, creates prognostic uncertainty and can make decisions about redirection of care towards palliation more stressful than they already are (Table 5: Quote 2). They explained that some patients are so young in the NICU that they do not always present with observable features of a life-limiting condition, which can make it more difficult for both HPs and families to appreciate that the prognosis is likely to be poor (Table 5: Quote 3). 

When exploring this view in the focus groups, HPs discussed that despite having extensive experience in caring for critically ill newborns, medical teams from the NICU/PICU have little experience managing older patients with the same diagnosis and emphasized how urGS testing can bridge the gap between HPs from pediatrics and adult populations, as well as across departments (Table 5: Quote 4). Participants explained that reaching out to non-NICU/PICU specialty teams for advice and insights on the management of older patients with the same diagnosis can aid decision making regarding the best treatment pathway for a NICU/PICU patient. 

## 4. Discussion

Utilization of urGS in the NICU/PICU setting is challenging because it occurs at a time when new parents are overwhelmed, and the implications of the genomic information provided may be unclear. The short timeframe in which testing takes place means any diagnosis, especially where palliative care for an infant is considered, needs to be processed fairly quickly by both the HPs and the infant’s family. Our findings are important, timely, and reveal the complex nature of using urGS in the NICU/PICU. 

Our findings revealed that some HPs prioritize urGS over other tests when trying to diagnose a condition, and this contrasts with earlier research by Lynch et al. who reported that HPs in the NICU/PICU focused on other tests rather than urGS [28]. This difference may reflect the NICU/PICU workforce becoming more familiar and confident with the technology as they continue to witness diagnoses being made in timeframes that were not previously possible. HPs may prioritize the urGS test result because they see it as an efficient use of resources in a time-sensitive environment [6,29,30]. Strict monitoring of patient eligibility will need to remain in place to maintain good stewardship of the resource. The promotion of integrated multi-disciplinary input into patient care and training of the wider NICU/PICU workforce on the interpretation of urGS results is also critical to prevent unnecessary use of urGS, and to ensure adequate pre-test counseling and results are clearly explained to families and all staff [31].

Obtaining the urGS test result in the NICU/PICU means that the decision of whether to redirect care towards palliation may happen earlier in the treatment pathway. There were two important elements related to the unique timing of the decision. First, the rapid nature of the test means that families receiving testing do not have time to reflect on the potential scope of results. Second, the predictive nature of the results can be even more devastating for families in NICU/PICU than in other settings because they are being disclosed so early, not only in the diagnostic journey but also in the parenting journey. A survey of parents whose children had urGS testing in the NICU/PICU also highlighted the convergence of these two elements as unique, explaining that it is important for HPs offering urGS to families to weigh the likely benefit of the test against the family’s ability to cope with the result at that point in time [32]. Our HPs discussed the need to assess a family’s ability to cope with different potential test outcomes, such as a diagnosis of a life-limiting condition or one associated with an intellectual disability. Determining family-specific support structures and an estimation of the parent’s ability to cope with the result prior to offering a urGS test in this setting may influence whether, how, and when to offer urGS. 

HPs in our study felt parents had insufficient understanding to provide informed consent to the urGS test, a concern supported by parents who were often confused by the words “screening” and “diagnostic” tests [33]. However, parental surveys conducted in this setting have also found the vast majority of parents feel they received enough information [14,32]; HPs may be presuming parents require greater understanding than they actually need in order to provide consent. Given the high stakes and stressful nature of the NICU/PICU, some authors have suggested that appropriately—rather than fully—informed consent should be the standard in this setting [34]. While for some parents, the knowledge that there is a chance the test can help their critically ill child may be sufficient for them to decide to undergo urGS, others may require more information before making a decision. 

Both genetically and non-genetically trained participants using urGS within the NICU/PICU explained that they have adapted their genetic counseling to alert parents more explicitly that redirection of care towards palliation may be presented to them post-urGS testing. Our findings support those of Lynch et al., who found that GCs modulate their pre-test counseling to prepare families to make difficult decisions [28]. This has important implications for training HPs who will be delivering pre-test counseling in the NICU/PICU. Genetic counselors working in acute care may, therefore, need additional training on discussing redirection of care towards palliation with families. 

Some HPs in our study expressed that the timing of the urGS test and delivery of the result may impact parent–child bonding, simply by conducting it so early in the parenting journey; parents seem to emotionally disconnect themselves from their child and avoid discussions about the next treatment steps once a diagnosis of a severe/life-limiting condition is delivered. Emotional disconnection of parents from their children, and the parents’ subsequent ability to make decisions, has been observed in other settings, such as prenatal diagnosis [35]. BabySeq used the Mother-to-Infant Bonding Scale to survey 60 families with newborns who had been admitted to intensive care and had a newborn genomic sequencing test. Of this group, almost half (29) of the families received a monogenic disease risk finding, and no significant increase in harm to their parent–child relationship was reported [12]. Yet, the authors did not distinguish between those families that did and did not receive a diagnosis of a life-limiting condition making it difficult to compare this research with our findings. In contrast, Bowman-Smart et al. found that families of children who had received a diagnosis through urGS reported reduced family functioning compared to those who did not receive a diagnosis and hypothesized that reduced family functioning may disrupt bonding [13]. As such, whether urGS impacts bonding in this setting remains unclear and warrants further exploration. 

Using urGS in the NICU/PICU can exacerbate conflicts between HPs and families about how to proceed with treatment options for a sick child. HPs cited examples where a diagnosis via urGS provided a viable treatment pathway but there was a chance the child would still have moderate intellectual disability. Parents’ decisions did not always align with the treating team: the parents requested redirection of care towards palliation despite HP’s advice that a treatment option was available (although the intellectual disability would remain). While families appeared to make decisions based on their concepts of intellectual disability, support structures, and expected quality of life, the HPs were perhaps more focused on survival and less on the challenges of raising a child with an intellectual disability. Our findings support those of Mills and Cortezzo, who found that misalignment in value judgment, coupled with the pressure of abiding by parental wishes, makes the decision process profoundly distressing for HPs in this setting [36]. The conflict between families and HPs is perhaps even more pronounced when using urGS in the NICU/PICU because of the limited time available for parents to become accustomed to the idea that their child is critically ill and what it might be like to have a child with a disability [37]. Implementation of formal workplace support structures, such as hospital clinical ethics committees, for all HPs dealing with the ethical challenges of this decision with families, is recommended. In addition, reminders to HPs that this is a possible response from parents may be helpful to manage HPs’ expectations for parental decision making regarding the redirection of care toward palliation.

In theory, identifying rare conditions is no different with urGS compared to other means. In many cases, a diagnosis of a rare condition provides closure and helpful guidance regarding treatment options. However, when a rare disease is diagnosed rapidly in infancy, as is the case with urGS, it can create uncertainty about the future due to a lack of knowledge about the disease [38]. Our results suggest that HPs feel uncomfortable suggesting redirection of care towards palliation when there is little evidence to support prognostication. They also found informal and opportunistic communication with other departments managing older patients with the same diagnosis helpful in treatment decision making. This suggests HPs may benefit from increased purposeful communication between non-intensive and NICU/PICU medical teams when reviewing specific patient groups to aid perceptions of what future care would look like in the NICU/PICU. This cross-pollination of knowledge could be an intentional collaboration of pediatric, adolescent, and adult health services or secondments and mentoring of staff between departments treating the same patient groups and conditions. 

This study has several limitations. Although we attempted to recruit widely across the ACG program, no nurses and only two intensivists participated. As such, our findings may not be representative of all HPs working in this setting. However, it is important to note that genetic services work across departments and the clinical geneticists and counselors interviewed for this study were recruited because of their role as part of the multi-disciplinary NICU/PICU teams within their health service. Our study sought to include HPs from the ACG program to ensure participants were experienced in using urGS in the NICU/PICU setting but this resulted in a geographical bias which reflects the composition of the genomics workforce and the number of state-based referrals [39]. Therefore, these findings may not be extrapolatable to other widely distributed urGS programs. 

## 5. Conclusions

Our findings have several important implications for the use of urGS in clinical practice. Despite urGS raising ethical questions, it has previously been shown to be invaluable to families. Our study highlights a need for more education for non-genetic HPs, especially in the NICU/PICU about the limitations of urGS and the importance of multi-disciplinary input and intentional communication and cross-pollination of knowledge between non-intensive and NICU/PICU medical teams when making treatment decisions. Our findings also suggest the optimal time for offering urGS and assessment of each family’s ability to cope with a life-limiting diagnosis should be routinely considered, allowing parents the chance to adjust and make sense of the possible outcomes for their child and minimize the impact of urGS on parent–child bonding. Participants described how they adapt their pre-test counseling in order to discuss redirection of care towards palliation more readily with families. As a result, guidelines for HPs conducting pre-test counseling in this setting may need to be developed to ensure parents are well prepared to receive results with poor outcomes and are adequately warned that redirection of care towards palliation may be actively considered. Finally, our findings suggest that conflict between families and HPs is even more pronounced when using urGS in the NICU/PICU because of the limited time available for parents to become accustomed to the idea that their child is critically ill. Further development of ways to manage individual families, perhaps on a case-by-case basis, and the use of formal workplace support structures, such as ethics committees, may assist and support HPs through the decision-making process in this setting.

## Figures and Tables

**Table 1 children-10-00824-t001:** Illustrative quotes for prioritization of the urGS results category.

Sub-Categories	Illustrative Quotes
Consideration of the entire clinical picture dependence on urGS results for decision making	*Quote 1: “My observation is that, with all intensivists and all investigations, there is a risk that we become too dependent on getting a result. (…) we are often [trying to] get answers, to try and put the pieces of the puzzle together and I worry that sometimes that we want results for our sake instead of the family’s sake.” FG_Int_2*
Dependence on urGS results for decision making	*Quote 2: “In terms of reliance on genomic results for decision-making, and even in negative cases, or cases where we haven’t got a diagnosis, we (…) push very hard back to the NICU in terms of they should be making a clinical decision based on the clinical picture of the child. Not relying on genetic results. (…) these results can help give evidence about prognosis, they can help guide those but still they need to look at the clinical picture or look at the baby or child.” FG_ClinG_1*
HPs overstate the chances of getting a diagnosis	*Quote 3: “NICU/PICU (…) are saying: “Well we are going to do a test that will tell us what to do next, it will tell us what’s happening with your child, it will tell us how to manage them”.” FG_P_2*
Lack of understanding by non-genetics HPs	*Quote 4: “I’ve had (…) NICU and cardiologists who have said: “Can we do the fancy test? Can we do the fast test?” (…) So, their technical understanding of what’s going on is insufficient to kind of understand the complexities (...). I feel there is a big challenge of just, education (...). But there are definitely some colleagues who have insufficient understanding of what’s actually involved and they think: “Just do the fancy test”.” FG_ClinG_3*

**Table 2 children-10-00824-t002:** Illustrative quotes for the timing of the urGS test category.

Sub-Categories	Illustrative Quotes
Bonding was disrupted due to conducting the urGS test and receiving a result	*Quote 1: “...sometimes a diagnosis has cemented a family’s decision earlier than what may have been made prior. I think without a diagnosis (…) families would take some time to see how things would pan out a little bit longer. Sometimes a diagnosis may cement their ideas about that, the future that they don’t want to accept. So, there’s less opportunity for bonding, for attachment, for, potentially forming a relationship that might have changed decisions later on in life.” FG_Int_2*
Parents disengage from having discussions	*Quote 2: “A baby was diagnosed [through urGS], and the outcome for kids with this condition is usually dire, and so the parents were advised of their options. One option was palliation, which is what [the parents] chose to do. At the time the parents were very distraught. (…) Dad had not even looked at the baby, (…) because he already loved that child so much that to form a stronger bond would have broken him. (…) The medical team were wrong regarding the outcome, for this baby. (...) he is still alive, 18 months later, when everyone thought it would be weeks, (…) and he would die.” FG_GC_5*
More time to bond may result in different outcomes	*Quote 3: “One of those families had got the diagnosis when the child was nine and had gone through a non-rapid exome after many, many years of searching and (…) the mother was so relieved, teary, but in a relieved, “how amazing is it that we know what this is!” (way).* *And I (...) contrasted it with this child who was 5 days old, and they came into hospital (had a urGS test) and then, got the diagnosis (...) three or four days after that. And (…), they hadn’t had years of this child being just who they were. They had this [reaction] “this is the new baby, and this is what we’re supposed to cope with?”” FG_GC_7*

**Table 3 children-10-00824-t003:** Illustrative quotes for adaptation of pre-test counseling category.

Sub-Categories	Illustrative Quotes
Adaptation of pre-test counseling	*Quote 1: “… we say to the parents, “look we might have an explanation for what we are seeing right now, but we might also be coming back to you with some difficult information about long term prognosis”. And I think we have to plant that seed early…” I_ClinG_1* *Quote 2: “I used to bring it up but not in a very directive way, (…) but [now] I just [plant] a seed in their mind that we might be talking about something horrific to hear and there might be decisions made about palliation based on this result which is certainly how I would approach it now.” FG_GC_2*
Highlighting potential dire outcomes	*Quote 3: “I was more worried that a severe intellectual disability would turn up as well. So, I said, “So this might show you something that you really didn’t expect, including (…) severe developmental outcomes.” FG_ClinG_5*
Early awareness appears to help HPs in later discussions	*Quote 4: “I think it makes a difference when you are seeing people for result giving.” FG_GC_2*

**Table 4 children-10-00824-t004:** Illustrative quotes for the misalignment between HPs and families category.

Sub-Categories	Illustrative Quotes
Discomfort ending a life	*Quote 1: “The ones that stand out as morally distressing are the ones where there is a disconnect between the family decision-making and what (…) the medical team think. One that really stands out for me, (…) was a diagnosis of a metabolic condition that was treatable, with enzyme therapy. So, it’s not an easy treatment, and [you] don’t quite get a perfect child at the end of it. (...) but the family decided to withdraw care. The clinicians that usually look after those conditions, felt quite distressed by that. Because in their experience, although the outlook is not of normal life, it’s a typical output of treatment.” FG_ClinG_5*
Fear of parental decision regret	*Quote 2: “There are quite a few families that I’ve seen that are, well, you would be damaged. But I think that, we’ll it’s hard to say, but my feeling is that the decisions they’ve made are decisions they are going to regret in the future.” FG_GC_5*
Decision within the zone of parental discretion	*Quote 3: “It’s more the mild distress of dealing with situations of that perhaps families are choosing choices that are a little uncomfortable to you but still fit within the zone of parental discretion. So, its (...) accepting decisions that perhaps are sitting towards the outer edge of what we feel comfortable with, even though we may as a team consider that its ethically permissible, per se.” FG_Int_ 2* *Quote 4: ClinG_5: “[The] family just decided, (…) [it] wasn’t really the long-term plan for them (…). Not really what they had in mind when they had the baby. Um, so they decided to let the baby go(...)* *Facilitator: And what did you find the most challenging about that case?* *ClinG_5: Having other doctors in my office distressed by this outcome (…). It’s not a condition that causes severe intellectual disability. Ultimately, it’s the family decision, because (…) they have to look after the child and live with the child, so we [the medical team] all felt that this was an informed decision. It just went against what we were expecting.” FG_ClinG_5*

**Table 5 children-10-00824-t005:** Illustrative quotes for the prognostic uncertainty category.

Sub-Categories	Illustrative Quotes
Up-to-date prognostic information is needed	*Quote 1: “I was involved with another family where a [life-limiting/severe] diagnosis was made. [However] the course that child has had was very different from the clinical information provided to the family, because presumably that [clinical information] was based on children who didn’t get an early diagnosis and were damaged because of the condition [and lack of appropriate intervention]. Whereas this child is now doing extremely well. (...) the information that we gave [the family] was wrong. It was based on outdated information.” FG_GC_9*
Lack of evidence in the literature for guiding decisions for rare conditions	*Quote 2: “There are only two cases in the literature about children with variants in this gene and its variable in terms of the severity of the intellectual disability, very severe in some individuals but then there is someone else who was 12 and was riding a bike! So incredibly difficult to make those types of decisions.” FG_ClinG_2*
Lack of phenotypic evidence in newborns to support decision making	*Quote 3: “I work with people who have been seeing families in the outpatient setting, they are very experienced, for 20 years they recognise intellectual disability in a different way. I guess when people are 10 or 15 years of age, [HPs] understand what intellectual disability means. But the experience they are lacking (…) is recognising what a bad outlook looks like in a brand-new baby that doesn’t walk doesn’t talk yet. Not relying on the same sort of things that would usually tell you that they are not going to do well. And that’s been quite confronting, because some of the time, they just look normal (…) if you were to go and look at [the baby], they look okay.” FG_ClinG_5*
Multi-disciplinary input can influence decision making	*Quote 4: “…once we had the diagnosis, we knew the facts of the condition. The respiratory team who was involved said, “We have a couple of these patients who we see in out-patients, this is what their prognosis has been, they come off the ventilators.” (…) [This input] actually flipped it the other way, to say, “No we should actually advocate for active care for this child. Yes, they are not going very well now, and there’s a general poor prognosis, but we also know that they do very well once they get over this acute period”.” I_ClinG_1*

## Data Availability

Full data are not publicly available as participants consented to share relevant quotes and not full interview transcripts with third parties.

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
