# Peer review of "Rapid Genomic Testing in Intensive Care: Health Professionals’ Perspectives on Ethical Challenges"

_children, 2023, doi:10.3390/children10050824_

Round 1
Reviewer 1 Report
The main question addressed from this research is related to the perception and ethical challenges of rapid genomic testing introduction from Health care professional point of view the topic is original and relevant in the field definitively address a gap in the fieldthe introduction of such tests in routine practice with a quick result is changing completely the approach to genetic syndromes within the newborn and pediatric population but counseling is becoming every day more challenging for health care providers
I would suggest minor revisions - line 195-198 of the discussion I think have to be deletes - line 244 I would stress a bit the concept of appropriate counsel family regarding the meaning of each possible test, families often got confused from words "screening" / "diagnostic" tests (with this purpose I would suggest to cite PMID: 33111167) - I would suggest to reduce in size the discussion it is too long the reader may loose the sense conclusions are consistent refrences appropriate I would suggest to add a table with a summary (really brief sentence ) with a description of the genetic tests argued trough the paper and the evidence from your research best regards
Author Response
- Line 195-198 of the discussion I think have to be deletes.
We have deleted the sentences as requested.
- Line 244 I would stress a bit the concept of appropriate counsel family regarding the meaning of each possible test, families often got confused from words "screening" / "diagnostic" tests (with this purpose I would suggest to cite PMID: 33111167).
We have added a sentence to the discussion to address this point (please see additional text below) and cited the paper suggested.
“HPs in our study felt parents had insufficient understanding to provide informed consent to the urGS test, a concern supported by parents who were often confused by the words "screening" and "diagnostic" tests [32].”
- I would suggest to reduce in size the discussion it is too long the reader may loose the sense.
The discussion has been reduced in size where possible by simplifying the language used and by removing any sentences that appeared to be a duplication of wording. For ease of reading please see the clean revised version of the manuscript.
- I would suggest to add a table with a summary (really brief sentence ) with a description of the genetic tests argued trough the paper and the evidence from your research.
We apologise but we are not sure which tests reviewer #1 is referring to. However, to clarify, the purpose of the paper is not to argue for any specific tests but to outline the ethical challenges experienced by those who are using rapid genomic sequencing within acute care settings.
Reviewer 2 Report
Dear Authors
Your article is very interesting
1. In your article, it must emphasised the usefulness of this method in the diagnosis of very serious conditions.
As an example, Donahue syndrome could be diagnosed within hours.
2. Thus, despite ethical challenges of this method, the “undisputed” information for serious conditions could be very valuable for the parents.
3. Some language “errors” should be corrected. Examples: Incorrect spaces in lines 16 and 18.
Author Response
- In your article, it must emphasised the usefulness of this method in the diagnosis of very serious conditions. As an example, Donahue syndrome could be diagnosed within hours.
We agree that the usefulness of this method in diagnosing serious conditions is important to note. However, this research was not aimed at exploring the utility of urGS as this has already been established and we have provided references to support this. To address your comment, we have added the following statements to the introduction:
“Diagnoses achieved through urGS can be used to guide clinical management of serious conditions, enable access to life-saving treatment (in some cases within a matter of hours), and provide informed reproductive options for families [1-5].”
- Thus, despite ethical challenges of this method, the “undisputed” information for serious conditions could be very valuable for the parents.
We have added the following additional text to two sections of the manuscript to emphasise that the information is considered valuable for parents:
Discussion: “However, parental surveys conducted in this setting have also found the vast majority of parents feel they received enough information and that the testing was somewhat useful [14, 31];”
Conclusion: “Despite urGS raising ethical questions it has previously been shown to be invaluable to families...”
- Some language “errors” should be corrected. Examples: Incorrect spaces in lines 16 and 18.
We have corrected the language errors throughout.
Reviewer 3 Report
The work is devoted to the study of problems arising from the use of urGS in intensive care units. Based on the analysis of the discussion in focus groups, the authors formulated 4 major problems of using these methods.
This is an important result, as it can be used to train specialists working in intensive care units.
The work is devoted to the study of problems arising from the use of ultra-high-speed genome sequencing in intensive care units. Based on the analysis of the discussion in focus groups, the authors formulated 4 major problems of using these methods. This is an important result, as it can be used to train specialists working in intensive care units.
The content of all sections of the article is quite satisfactory. The discussion is written in great detail and partially duplicates the results. From the point of view of design, it is necessary to remove the technical comments of lines 102-104 and 195-198.
Remarks
Disadvantages of the work
1. The authors write that they investigated the work of the staff of intensive care units. However, 17 of the 19 study participants are geneticists. At the same time, the formulated problems of using the method concern the staff of intensive care units. How can the results of the study conducted with such focus groups be extrapolated to the staff of intensive care units?
2. The principles of dividing focus groups into groups of 5 specialists are not clearly described. After all, the problems discussed depend on the composition of the group.
In my opinion, this paper formulates important issues that need to be paid attention to when training specialists. However, the scientific way of obtaining these results is not entirely clear.
Author Response
- The discussion is written in great detail and partially duplicates the results
We have attempted to reduce the duplication of the results and condense the discussion while still retaining the important content.
- From the point of view of design, it is necessary to remove the technical comments of lines 102-104 and 195-198.
These two sections have been removed.
- The authors write that they investigated the work of the staff of intensive care units. However, 17 of the 19 study participants are geneticists. At the same time, the formulated problems of using the method concern the staff of intensive care units. How can the results of the study conducted with such focus groups be extrapolated to the staff of intensive care units?
This is because the genetic health professionals interviewed were working within intensive care units in order to deliver rapid genomic testing as a service. We have added a sentence to the limitations section to make this clearer, which now reads:
“However, it is important to note that genetic services work across departments and the clinical geneticists and counsellors interviewed for this study were recruited because of their role as part of the multi-disciplinary NICU/PICU teams within their health service.”
- The principles of dividing focus groups into groups of 5 specialists are not clearly described. After all, the problems discussed depend on the composition of the group.
We have added the following sentences to the methods to address this point, which now reads “Focus group size was deliberately restricted to a maximum of five participants and two facilitators (KA and DV) to allow each participant’s views to be heard within the time allocated and to comfortably manage the session online. Focus groups were allocated taking into account profession and level of seniority wherever possible.”
- However, the scientific way of obtaining these results is not entirely clear.
We hope that the response to comment 3 above clarifies this point.
Round 2
Reviewer 3 Report
The authors of the manuscript explained the doubtful points in the answer and included them in the text of the article. Now the principle of forming groups and studying the issue has become clear. Technical edits have also been made to the manuscript. I consider this work worthy of publication.